# OpenReview forum: "Adaptive Reinforcement Learning for Unobservable Random Delays"
_ICML.cc/2026/Conference — ICML 2026 regular_

### Official Review · Reviewer_mLrX · 2026-03-03

**Soundness:** 3
**Presentation:** 2
**Significance:** 2
**Originality:** 3
**Overall Recommendation:** 4
**Confidence:** 4

**Summary:**

In this paper, authors come up with a systematic approach to solve the unobservable random delay problem for delayed RL. The approach is clearly illustrated with intuitive figure demo and the experiments are comprehensive. Specifically, it introduces an interesting POMDP formulation together with a variant of Dreamer-like MBRL algo to tackle the random delay problem under the RL setting.

**Compliance With Llm Reviewing Policy:**

Affirmed.

**Final Justification:**

I remain positive for the following paper.

**Key Questions For Authors:**

1, For the choice of latent representation for actual state distributions and dynamics is it possible to try something different? Like diffusion or transformer? \
2, Is it possible to simplify the design of action packet and action buffer? Or is it possible to combine these two components into a single component?\

**Limitations:**

1, Current empirical studies are conducted on fairly easy mujoco locomotion tasks. It would be better to extend it to more complicated manipulation/dexterity robotics tasks or other more challenging tasks.\
2, Comparison with current SOTA belief/model-based methods are needed such as [1].\
3, It would be better to have a system-level algorithmic flowchart to help reader keep track of algorithmic progression, since current system is bit too complicated to understand by pure text.\
4, The paper claims to address unobservable random delays. However, because the system includes a global timestamp for each state/action packet in the interaction layer, these delays can actually be derived from the recorded timestamps. Therefore, describing them as “unobservable” is an overstatement; a more accurate term would simply be “random delays.”\
5, Citation missed for the belief/model-based method [2].\
6, I would recommend author to provide a more comprehensive system overview diagram, since current approach is pretty complex with extensive use of parameters and symbols for different data.


[1]: Wu, Qingyuan, et al. "Directly Forecasting Belief for Reinforcement Learning with Delays." International Conference on Machine Learning. PMLR, 2025.\
[2]:Zhan, Simon Sinong, et al. "Belief-Based Offline Reinforcement Learning for Delay-Robust Policy Optimization." The Fourteenth International Conference on Learning Representations.

**Strengths And Weaknesses:**

The problem setting itself is meaningful since current delayed RL still stuck with random delay setting. The algorithm is sound and solid borrowing some real-time system thoughts in solving Delayed RL problem, which is novel for the overall RL context. Though setting and assumptions are bit different from current delayed RL, work itself is solid with some insightful thoughts. In addition, the empirical study is comprehensive and backing up the statement of the authors. However, even though author claim they solve the unobservable delay setup, they assume quite strong assumptions on global timing, etc., which lower down the overall significance of the paper.

---

> ### Author Rebuttal · Authors · 2026-03-31
>
> Thank you for reviewing our paper. We address your questions and concerns below, and we are also willing to engage in further discussion if needed.
>
> > However, even though author claim they solve the unobservable delay setup, they assume quite strong assumptions on global timing, etc., which lower down the overall significance of the paper.
>
> Thank you for your comment, which actually highlights an important contribution of our paper: We do not make any assumption about global timing. The time tag is local to the interaction layer only, and the agent itself has no concept of time. The agent decides what could happen if an action packet arrives with a certain round-trip delay, but never attempts to measure the delay itself.
>
> We have tried to explain this in a dedicated paragraph on lines 258-262 in the paper, but we now plan to extend the discussion. In combination with your suggested improvement to the system diagram, we will update Figure 2 to make the concept of non-global time even clearer.
>
> > For the choice of latent representation for actual state distributions and dynamics is it possible to try something different? Like diffusion or transformer?
>
> The current GRU-based model architecture was primarily chosen for its interpretability: an embedded distribution is updated when an action is applied (Appendix D.1). A transformer would be a viable alternative, albeit requiring more complex configuration than our GRU-based model. We are unsure how a diffusion model would work for our problem, as it does not naturally encode variable-length sequences. We did not explore these two models further as our model showed acceptable performance under constant delay (Appendix E.3).
>
> > Current empirical studies are conducted on fairly easy mujoco locomotion tasks. It would be better to extend it to more complicated manipulation/dexterity robotics tasks or other more challenging tasks.
>
> We chose the MuJoCo environments because they are challenging from a delay perspective. This is the standard choice for delayed MDP benchmarks, also used by both [1] and [2]. Manipulation tasks, while challenging in general, are less sensitive to interaction delays since the object being manipulated is often either static or held by the arm. Locomotion tasks, such as a humanoid walking, experience balancing issues that require reactive corrections, where delays directly impact reaction time.
>
> > Is it possible to simplify the design of action packet and action buffer? Or is it possible to combine these two components into a single component?\
>
> The action buffer is an active component at the interaction layer that exists to cover gaps in interactions, i.e., time steps when no new action packet arrives. Each row in an action packet specifies the new buffer contents to apply if the packet arrives with a certain round-trip delay (measured by the interaction layer). In our view, this is the simplest design that still preserves handling of interaction gaps and variable-length unobservable delays. However, there is room for optimizations to reduce computational and communication overhead, as discussed in Appendix F.
>
> > Comparison with current SOTA belief/model-based methods are needed such as [1].
>
> > Citation missed for the belief/model-based method [2].
>
> VDPO, a baseline we compare against, is similar to [1] in its belief method. We are aware of the work in both [1] and [2], but did not include them in our evaluation since they are offline RL methods and therefore out of scope. Offline RL assumes the existence of a dataset with demonstrated behavior, whereas ACDA directly learns a policy by exploring the delayed environment. Although we cite [1] in the related work section, we will update the paper to make this distinction even clearer.
>
> Thank you again for your review and your suggestions for improvements! We hope that we have addressed your concerns and are happy to engage in further discussion.

---

> > ### Author Rebuttal · Reviewer_mLrX · 2026-04-02
> >
> > Author's rebuttal has solved majority of my questions. I don't have any further question till this point.

---

### Official Review · Reviewer_ULjq · 2026-03-09

**Soundness:** 4
**Presentation:** 3
**Significance:** 3
**Originality:** 3
**Overall Recommendation:** 5
**Confidence:** 2

**Summary:**

This paper addresses the challenge of unobservable random interaction delays in reinforcement learning,  which is a common issue in real-world cyber-physical systems. The authors propose a novel framework consisting of an "Interaction Layer" and the Actor-Critic with Delay Adaptation algorithm.

**Compliance With Llm Reviewing Policy:**

Affirmed.

**Key Questions For Authors:**

1. Can the authors provide a theoretical discussion on algorithm’s convergence properties and optimality?

2. The Appendix seems too long. I suggest the authors reorganize the Appendix by using a clearer hierarchical structure, providing a table of contents for the supplementary material.

**Limitations:**

yes

**Strengths And Weaknesses:**

Strengths
1. The experiments are comprehensive, covering various delay distributions and including real-world network delay traces (WiFi), which enhances the credibility of the results.

Weaknesses

2. While the proposed ACDA algorithm demonstrates strong empirical performance, the paper lacks a rigorous theoretical analysis regarding its convergence properties and optimality.

---

> ### Author Rebuttal · Authors · 2026-03-31
>
> Thank you for reviewing our paper. We address your questions and concerns below, and we are also willing to engage in further discussion.
>
> > While the proposed ACDA algorithm demonstrates strong empirical performance, the paper lacks a rigorous theoretical analysis regarding its convergence properties and optimality.
>
> > Can the authors provide a theoretical discussion on algorithm’s convergence properties and optimality?
>
> This is a good suggestion, and we will update the paper to include a discussion on the convergence and optimality of ACDA. In general, the convergence of deep RL algorithms such as SAC remains poorly understood even without any delays. Existing convergence results are established only in the tabular case [1]. To our knowledge, no prior work on stochastic delay frameworks analyzes convergence or optimality. Further complicating any theoretical analysis, ACDA explicitly targets non-i.i.d. delay distributions. There are special cases where existing MDP results apply, such as constant delays, and we will include this in the theoretical discussion.
>
> > The Appendix seems too long. I suggest the authors reorganize the Appendix by using a clearer hierarchical structure, providing a table of contents for the supplementary material.
>
> We do provide an outline of the appendices section describing the structure in text, but your suggestion of a table of contents is good, and we will update the paper to include it at the beginning of the appendices.
>
> In the current structure, the appendices are ordered by perceived importance:
>
> - Appendix A-B: Important details for the evaluation and methodology justification.
> - Appendix C-D: Interaction layer and ACDA expanded definitions.
> - Appendix E: Complete results.
> - Appendix F: Discussion about possible limitations of our work.
> - Appendix G: Practical implementation details.
>
> We consider all these appendices to be genuine contributions to the paper, although we agree that their contributions may not be clear from the current text in the outline of the appendices. We will update the paper to make this clearer and more direct.
>
> [1] Tuomas Haarnoja, Aurick Zhou, Pieter Abbeel, Sergey Levine. "Soft actor-critic: Off-policy maximum entropy deep reinforcement learning with a stochastic actor." In Proceedings of the 35th International Conference on Machine Learning, 2018.

---

> > ### Author Rebuttal · Reviewer_ULjq · 2026-04-01
> >
> > Thank you for your detailed response and for directly addressing my concerns during the rebuttal phase.
> >
> > Regarding the theoretical analysis, I appreciate your candid assessment of the landscape. Your proposed addition of a theoretical discussion section is a very reasonable approach.
> >
> > Thank you also for addressing the organization of the Appendix. Adding a Table of Contents at the beginning and explicitly clarifying the hierarchical structure and purpose of each section will greatly enhance the overall readability and navigability of your work.
> >
> > In light of these clarifications, I am happy to increase my score

---

### Official Review · Reviewer_Lt2A · 2026-03-09

**Soundness:** 3
**Presentation:** 3
**Significance:** 4
**Originality:** 3
**Overall Recommendation:** 5
**Confidence:** 2

**Summary:**

This article addresses the key issue of unobservable, time-varying delays in real-world reinforcement learning (RL) by proposing an ``interaction layer'' and the Actor-Critic with Delay Adaptation (ACDA) algorithm. Overall, this article's key findings constitute ACDA's strong ability to dynamically adapt to stochastic delays, significantly outperforming SOTA baselines in both simulated and real-world WiFi environments.

**Compliance With Llm Reviewing Policy:**

Affirmed.

**Final Justification:**

I am still not convinced that the case of i.i.d delays can be omitted.

**Key Questions For Authors:**

1.	How sensitive is the algorithm to the choice of prediction length $L$ and action buffer range $h$? How would the system perform if actual delays frequently exceed the predefined $L$?


2.	How robust is your heuristic assumption under completely uncorrelated delay distributions?

**Limitations:**

Yes

**Strengths And Weaknesses:**

Strengths:
1. Highly Practical Motivation: The problem of unobservable, stochastic delays is a major bottleneck for deploying RL in real-world robotics and networked systems. The proposed interaction layer is a very realistic and implementable system architecture.
2. Strong Empirical Results: The evaluation is thorough, utilizing both simulated queueing delays and actual measured WiFi delay datasets. ACDA demonstrates a clear and significant performance margin over existing constant-delay and random-delay baselines.

Weaknesses:
1. Reliance on Heuristic Assumptions: When inferring past executed actions, the algorithm assumes recent delays are constant. This works well with time-correlated delays but may degrade performance when delays are completely random and irregular.
2. High Overhead: Generating and transmitting the action matrix increases computation time and network bandwidth requirements. This may limit its application in high-frequency control or low-bandwidth scenarios.

---

> ### Author Rebuttal · Authors · 2026-03-31
>
> Thank you for reviewing our paper and for appreciating the real-world relevance and significance of our results. We address your questions and concerns below, and we are also willing to engage in further discussion.
>
> > Reliance on Heuristic Assumptions: When inferring past executed actions, the algorithm assumes recent delays are constant. This works well with time-correlated delays but may degrade performance when delays are completely random and irregular.
>
> We designed the heuristic to operate under realistic delay processes, where delays are temporally correlated rather than i.i.d. The measured WiFi delays are used to validate the merit of our temporally correlated delay assumption, and we further evaluate the performance of algorithms on M/M/1 and Gilbert-Elliott delays for reproducibility and interpretability. We expect that ACDA would not perform well under i.i.d. delays, but we are also unaware of a realistic deployment scenario where interaction delays are truly i.i.d.
>
> > High Overhead: Generating and transmitting the action matrix increases computation time and network bandwidth requirements. This may limit its application in high-frequency control or low-bandwidth scenarios.
>
> This is indeed true, although our analysis in Appendix F shows that this overhead is not a problem in the evaluated MuJoCo environments. We also propose and evaluate various mitigations should the overhead become a problem, such as an action-skipping approach that maintains acceptable performance (Appendix F.4). We will add a dedicated limitations section in the main body summarizing these considerations, as described in our response to reviewer cJTG.
>
> > How sensitive is the algorithm to the choice of prediction length $L$ and action buffer range $h$? How would the system perform if actual delays frequently exceed the predefined $L$?
>
> The evaluated M/M/1 delay occasionally exceeds the predefined prediction length $L$. On average, 1 in every 1000 packets will exceed $L$. When this happens, that packet with $d > L$ is discarded. This process is described on lines 168-178 and 1308-1313 in the paper. We did not attempt to optimize $L$, as there is no downside to a larger $L$ provided that bandwidth and computational overhead remain acceptable.
>
> As stated on lines 165-166 in the paper, the horizon $h$ exists only to cover gaps in the interaction. Too short a horizon means that the last received action will be repeated (Figure 4). This effect is present regardless of the algorithm. While we never experimented with a too-short horizon $h$, this action-repetition behavior is captured in the ACDA training process using information from $c_t$.

---

> > ### Author Rebuttal · Reviewer_Lt2A · 2026-04-04
> >
> > Thanks for providing the response. I am not convinced that the case of i.i.d delays is not considered. I do hope the author can address this kind of delay.

---

> > > ### Author Response · Authors · 2026-04-07
> > >
> > > Thank you for your acknowledgement. Please allow us to expand on our previous remarks regarding i.i.d. delays.
> > >
> > > Our position is that i.i.d. delays do not reflect real-world deployment conditions. Both our own measurements and those from prior work [1, 2] show that network delays exhibit a strong temporal correlation, with the delay concentration shifting over time. This is also the scenario for which the ACDA heuristic was constructed.
> > >
> > > However, we want to clarify that nothing in the ACDA design prevents it from operating under i.i.d. delays. While the heuristic assumes that consecutive delays are similar, this remains a decent approximation even for i.i.d. distributions where samples are likely to cluster on the mean.
> > >
> > > The interaction layer is also a general architectural contribution that extends beyond ACDA's heuristics. For example, by arranging the contents of action packets in a specific way, the interaction layer allows existing constant-delay algorithms to be applied in random-delay settings, as demonstrated in Appendix A.
> > >
> > > We agree that evaluating ACDA under i.i.d. delays is of interest, and we have therefore run benchmarks on the i.i.d. WiFi histogram delays from [3]. These results will be included as an appendix in the final paper, with a clear reference in the main text.
> > >
> > > Preliminary results are presented in the two tables below. The first table presents results when replaying real-world WiFi delays, and the second table presents results when sampling delays i.i.d. from a histogram.
> > >
> > > **Results when replaying real-world measured WiFi delays (DLib dataset in our paper):**
> > >
> > > | | Ant-v4 | Humanoid-v4 | HalfCheetah-v4 | Hopper-v4 | Walker2d-v4 |
> > > |-|-:|-:|-:|-:|-:|
> > > | SAC /w CDA | 681.24 | 595.15 | 232.49 | 388.19 | 481.83 |
> > > | BPQL | 2423.44 | 938.63 | 3209.00 | 2824.50 | 994.69 |
> > > | ACDA | 4381.00 | 5605.52 | 8368.57 | 3202.64 | 4424.82 |
> > >
> > > **Results from i.i.d. delays sampled from the WiFi histogram [3]:**
> > >
> > > | | Ant-v4 | Humanoid-v4 | HalfCheetah-v4 | Hopper-v4 | Walker2d-v4 |
> > > |-|-:|-:|-:|-:|-:|
> > > | SAC /w CDA | 431.90 | 733.40 | 977.76 | 2846.49 | 1670.96 |
> > > | BPQL | 2973.94 | 5527.98 | 4993.56 | 2885.95 | 3550.99 |
> > > | ACDA | 2616.68 | 4015.82 | 3955.66 | 3189.10 | 4187.74 |
> > >
> > > We see that ACDA is robust across the board, excelling at real-world delays while maintaining acceptable performance even under i.i.d. delays.
> > >
> > > [1] Himabindu Pucha, Ying Zhang, Z. Morley Mao, and Y. Charlie Hu. 2007. Understanding network delay changes caused by routing events. SIGMETRICS Perform. Eval. Rev. 35, 1 (June 2007), 73–84. https://doi.org/10.1145/1269899.1254891
> > >
> > > [2] Waleed Reda, Kirill Bogdanov, Alexandros Milolidakis, Hamid Ghasemirahni, Marco Chiesa, Gerald Q. Maguire, and Dejan Kostić. 2020. Path persistence in the cloud: A study of the effects of inter-region traffic engineering in a large cloud provider's network. SIGCOMM Comput. Commun. Rev. 50, 2 (April 2020), 11–23. https://doi.org/10.1145/3402413.3402416
> > >
> > > [3] Bouteiller, Y., Ramstedt, S., Beltrame, G., Pal, C., and Binas, J. Reinforcement learning with random delays. In International Conference on Learning Representations,
> > > 2021. URL https://openreview.net/forum?id=QFYnKlBJYR.

---

### Official Review · Reviewer_cJTG · 2026-03-13

**Soundness:** 3
**Presentation:** 3
**Significance:** 3
**Originality:** 3
**Overall Recommendation:** 4
**Confidence:** 2

**Summary:**

The paper studies reinforcement learning under stochastic and unobservable interaction delays, which frequently occur in real-world cyber-physical systems but violate the standard Markov Decision Process assumption of instantaneous action execution. To address this, the authors introduce the interaction layer, a framework that allows an agent to anticipate uncertain delays by generating a matrix of potential future actions, enabling robust behavior even when delays vary or action packets are lost.

Building on this framework, the paper proposes Actor-Critic with Delay Adaptation (ACDA), a model-based RL algorithm that dynamically adapts to delay patterns. Experiments on locomotion benchmarks with realistic delay processes show that ACDA outperforms existing delayed-RL methods, demonstrating improved robustness and performance in environments with stochastic interaction delays.

Key contributions:
- a novel framework, the interaction layer, which allows agents to adapt to randomly varying delays, even when these delays are unobservable
- a new model-based reinforcement learning algorithm, Actor-Critic with Delay Adaptation (ACDA), which leverages the interaction layer to adapt dynamically to varying delays
- evaluation of ACDA on a suite of MuJoCo locomotion tasks, using randomly sampled delay processes designed to mimic real-world latency sources and using recorded real-world delays collected from WiFi network communications.

**Compliance With Llm Reviewing Policy:**

Affirmed.

**Key Questions For Authors:**

-

**Limitations:**

I did not identify a separate discussion about the limitations. Regarding the societal impact, the only statement is "There are many potential societal consequences of our work, none which we feel must be specifically highlighted here."

**Strengths And Weaknesses:**

The paper appears technically sound, with a well-motivated problem formulation, review of related works, an appropriate methodological approach, and empirical validation on several benchmarks. The main paper is followed by an extensive appendix, with implementation and evaluation details, detailed model description, additional results of experiments. it seems that the main missing part is the description of limitations.

The presentation is good. I only identified some writing issues on p. 5 in the part:
"The heuristic assumes that, if a_t arrives at time t+k (it having delay k), then previous action packets will also have delay k. Such
that at−1 will arrive at time t + k − 1, at−2 at t + k − 2, etc."

It seems that the first sentence should be fixed (a_t instead of "it"?), and the second part should also be grammatically improved.

The paper seems to be significant as it seems to be the first of one of the first works that allows agents to make informed and controlled decisions under random unobservable delays in RL. Handling these delays robustly is critical for robotics and remote control systems, cyber-physical systems, etc. Therefore, the problem tackled by the paper is important and practically relevant.

Good originality, primarily through the introduction of a new conceptual framework (the interaction layer) for delayed RL, combined with an algorithmic instantiation (ACDA). The work builds on existing RL paradigms but offers a new perspective on how to model delayed agent–environment interactions.

---

> ### Author Rebuttal · Authors · 2026-03-31
>
> Thank you for reviewing our paper! We address your comments below, but are also happy to answer any questions you might have.
>
> > The paper seems to be significant as it seems to be the first of one of the first works that allows agents to make informed and controlled decisions under random unobservable delays in RL.
>
> Thanks for your comments on the novelty and relevance of our work. To our knowledge, this is the first framework that allows for controlled decisions under random, unobservable delays. In contrast, other random unobservable-delay frameworks, such as DCAC, do not control exactly when an action is applied.
>
> We also agree with all your feedback on grammatical improvements and will update the paper to address them.
>
> > I did not identify a separate discussion about the limitations. Regarding the societal impact, the only statement is "There are many potential societal consequences of our work, none which we feel must be specifically highlighted here."
>
> The limitations are primarily discussed throughout the paper (e.g., lines 256-257 and 270-274) and in Appendix F, which is dedicated to computational and communication delays and includes evaluation of potential solutions. We agree that a dedicated limitation section improves clarity, and we will update the paper to include it before the conclusion.

---

> > ### Author Rebuttal · Reviewer_cJTG · 2026-04-01
> >
> > Thank you for your response. For now, I have decided to keep my score.

---

### Decision · Program_Chairs · 2026-04-30

**Decision:**

Accept (regular)

**Comment:**

The paper studies RL problems with stochastic unobservable delay, representing state-observation delay, policy-computation delay and action-execution delay. The authors assume the existence of an interaction layer that observes the true delay. Their algorithm sends it a matrix of potential sequences of actions, each sequence corresponding to a different delay, and the interaction layer executes the sequence that corresponds to the true delay. For learning, the authors propose an actor-critic model-based approach (ACDA). A central assumption is that delays exhibit strong temporal correlation, and the algorithm therefore uses the heuristic that consecutive delays are identical. The method is evaluated in both synthetic environments (Gymnasium) and a real-world setting involving WiFi delay.

The reviewers found the problem setting well motivated and viewed the use of an interaction layer together with an action-sequence matrix as both novel and practically appealing. They also agreed that the empirical evaluation is thorough and provides convincing evidence of the proposed method’s effectiveness.

At the same time, the reviewers noted several weaknesses, including the communication overhead introduced by the approach and its strong reliance on temporal correlation in the delays. To address the correlation concern, the authors provided preliminary additional results with i.i.d. delays, showing acceptable performance when the correlation assumption is not met. The authors are encouraged to incorporate these results into the paper.

Another concern raised during the discussion was the assumption of perfect synchronization between the interaction layer and the environment; in practice, an additional, albeit smaller, delay may arise between them.

Altogether, the reviewers agreed that these limitations are relatively minor.

Therefore, I recommend accepting the paper.